# Analysis of Epidemiological Characteristics of Notifiable Diseases Reported in Children Aged 0–14 Years from 2008 to 2017 in Zhejiang Province, China

**DOI:** 10.3390/ijerph16020168

**Published:** 2019-01-09

**Authors:** Qinbao Lu, Zheyuan Ding, Chen Wu, Haocheng Wu, Junfen Lin

**Affiliations:** Department of Public Health Surveillance & Advisory, Zhejiang Provincial Center for Disease Control and Prevention, 3399 Binsheng Road, Binjiang District, Hangzhou 310051, China; zhyding@cdc.zj.cn (Z.D.); chenwu@cdc.zj.cn (C.W.); hchwu@cdc.zj.cn (H.W.)

**Keywords:** children, infectious diseases, epidemic characteristics

## Abstract

This study aims to learn the characteristics of morbidity and mortality of notifiable diseases reported in children aged 0–14 years in Zhejiang Province in 2008–2017. We collated data from the China Information System for Disease Control and Prevention in Zhejiang province between 1 January 2008 and 31 December 2017 of children aged 0–14 years. From 2008 to 2017, a total of 32 types and 1,994,740 cases of notifiable diseases were reported in children aged 0–14 years, including 266 deaths in Zhejiang Province. The annual average morbidity was 2502.87/100,000, and the annual average mortality was 0.33/100,000. Male morbidity was 2886.98/100,000, and female morbidity was 2072.16/100,000, with the male morbidity rate higher than the female morbidity rate (χ^2^ = 54,033.12, *p* < 0.01). No Class A infectious diseases were reported. The morbidity of Class B infectious diseases showed a downward trend, but that of Class C infectious diseases showed an upward trend. There were 72,041 cases in 22 kinds of Class B infectious disease and 138 death cases, with a morbidity rate of 90.39/100,000, and a mortality rate of 0.17/100,000. There were 1,922,699 cases in 10 kinds of Class C infectious disease and 128 death cases, with a morbidity rate of 2412.47/100,000, and a mortality rate of 0.16/100,000. The main high-prevalence diseases included hand-foot-and-mouth disease (1430.38/100,000), other infectious diarrheal diseases (721.40/100,000), mumps (168.83/100,000), and influenza (47.40/100,000). We should focus on the prevention and control of hand-foot and mouth disease, other infectious diarrheal diseases, mumps and influenza in children aged 0–14 years in Zhejiang Province. It is recommended to strengthen epidemic surveillance and undertake early prevention and control measures in order to reduce the younger children incidence rate of infectious diseases. Immunization planning vaccines can help achieve a significant preventive decline of infectious diseases.

## 1. Introduction

For a long time, children’s infectious diseases have been the number one disease type to harm children’s health and threaten children’s lives [1]. With the continuous development of medical undertakings, although human beings have made brilliant achievements in controlling and defeating children’s infectious diseases, the harm and threat of children’s infectious diseases are still very serious today [2]. Children’s infectious diseases are prone to various complications threatening children’s lives; therefore, understanding the occurrence and changes of children’s infectious diseases is of great significance to the prevention and treatment of infectious diseases and the promotion of children’s health.

Infection surveillance is important in infectious disease management and prevention. The surveillance of notifiable diseases in China was first initiated in the 1950s. Accurate and timely surveillance of infectious diseases laid the foundation for effective disease control and prevention in China. After the severe acute respiratory syndrome (SARS) crisis in 2003, the Chinese government strengthened the construction of the public health information system. China officially initiated the China Information System for Disease Control and Prevention (CISDCP) in January 2004. This system is the most comprehensive and macroscopic notifiable disease surveillance system in China [3]. Timely analysis of notifiable disease surveillance data to understand epidemic trends and their main characteristics is the basis for the prevention and control of infectious diseases.

Zhejiang province, located in the southeastern coast of China, has moist air, a mild climate, a developed economy, and large population mobility. It covers an area of 101,800 km^2^ and is one of the most densely populated provinces in China. By 2017, the population has reached up to 56 million, and the population aged 0–14 years is about 7.5 million.

In this paper, we described epidemiological characteristics of notifiable diseases in children aged 0–14 years reported in Zhejiang Province in 2008–2017, for the purpose of providing a reference for the prevention and control of infectious diseases in children in Zhejiang Province. The results are reported as follows.

## 2. Materials and Methods

### 2.1. Data Resources

The task of CISDCP was to passively collect and summarize data of notifiable diseases from all levels of hospitals in the entire country. The data were obtained from all the clinical and laboratory diagnosed cases in the CISDCP, with the symptoms onset date between 1 January 2008 and 31 December 2017 and the current living addresses of the subjects in Zhejiang Province, covering 39 types of notifiable infectious diseases. The population data of all children aged 0–14 years was obtained from the subsystem “Basic Information System” of the CISDCP.

### 2.2. Main Classification Indicators

Currently, 39 notifiable infectious diseases are included in CISDCP, classified as A, B and C according to their epidemic levels and potential population threats [4]. Both Class A and B notifiable diseases have a high risk of an outbreak with rapid spread, while Class C notifiable diseases cause less severe epidemics than those of Class B. Analysis and comparison were implemented according to different classification methods for notifiable diseases. The classification indicators were defined as follows.

#### 2.2.1. All Infectious Diseases Were Divided into 3 Categories

Class A (2 types: plague and cholera), Class B (26 types: SARS, AIDS, viral hepatitis, polio, human infection with avian influenza A (H5N1), measles, epidemic hemorrhagic fever, rabies, epidemic encephalitis B, dengue fever, anthrax, dysentery, tuberculosis, typhoid/paratyphoid, epidemic cerebrospinal meningitis, whooping cough, diphtheria, neonatal tetanus, scarlet fever, brucellosis, gonorrhea, syphilis, leptospirosis, schistosomiasis, malaria and human infection with avian influenza A (H7N9) [5], and Class C (11 types: influenza, mumps, rubella, acute hemorrhagic conjunctivitis, leprosy, epidemic and endemic typhus, kala-azar, echinococcosis, filariasis, other infectious diarrheal diseases, hand-foot-and-mouth disease [6]). According to the “Diagnostic criteria for infections diarrhea” (WS 271-2007), enacted by the Ministry of Health of the People’s Republic of China, “other infectious diarrhea” in CISDCP is defined as a group of infectious diarrhea diseases with pathogens including rotavirus, norovirus, salmonella, enteropathogenic *Escherichia coli* (EPEC), and enteroinvasive *Escherichia coli* (EIEC), but not including cholera, dysentery, and typhoid/paratyphoid. Influenza A (H1N1) was included in Class B infectious diseases on 30 April 2009 [7]. It was removed from Class B to Class C under management of existing influenza on 1 January 2014 [5].

#### 2.2.2. Class B Infectious Diseases Were Classified into Five Sub-Classes by Routes of Transmission, Etiology or Sources of Infection

These include intestinal infectious diseases (polio, hepatitis A, hepatitis E, hepatitis unclassified, dysentery, typhoid/paratyphoid), respiratory infectious diseases (SARS, measles, epidemic cerebrospinal meningitis, tuberculosis, scarlet fever, influenza A (H1N1), avian influenza A (H7N9), whooping cough, diphtheria), natural focal and insect-borne infectious diseases (epidemic hemorrhagic fever, avian influenza A (H5N1), Dengue fever, anthrax, rabies, epidemic encephalitis B, leptospirosis, brucellosis, schistosomiasis, malaria), blood-borne and sexually transmitted infectious infections (AIDS, hepatitis B, hepatitis C, hepatitis D, gonorrhea, syphilis), and neonatal tetanus. Among them, viral hepatitis was further classified according to their routes of transmission.

### 2.3. Data Analysis

Age groups were defined as 0–, 1–, 2–, 3–, 4–, 5–, 6–, 7–, 8–, 9–, and 10–14 years. The geographic distribution of reported notifiable diseases cases was determined on the basis of the 11 cities of Zhejiang Province.

The reported Morbidity (M) of reported cases was calculated by dividing the number of reported cases (C) via the CISDCP by the number of aged 0–14 years inhabitants (I) registered in local public health facilities (M = C/I). The mortality (M) was calculated by dividing the number of reported deaths (D) via the CISDCP by the number of aged 0–14 years inhabitants (I) registered in local public health facilities (M = D/I). Reported morbidity and mortality were calculated per 100,000 persons. The morbidity rates by male or female sex presented as per 100,000 of that sex category. Population density is calculated by dividing the number of inhabitants registered in public health facilities by land area in square kilometers. Cases density is calculated by dividing the number of reported cases by land area in square kilometers. Descriptive analysis method was used to analyze the overall morbidity, deaths and ranking orders of notifiable diseases in children aged 0–14 years in Zhejiang Province in 2008–2017. Chi-square test was used to compare count data. Pearson’s correlation test was used to analyze whether population density and cases density correlated with each other. A *p* < 0.05 was considered statistically significant. All analyses were performed using Statistical Package for the Social Sciences, version 16.0 (SPSS, Chicago, IL, USA), ArcGIS software (version 10.1, ESRI Inc., Redlands, CA, USA) and Excel 2016 (Microsoft, Redmond, WA, USA).

### 2.4. Ethical Statement

This study was exempt from institutional review board assessment. Firstly, policy documents and statistics data related to notifiable disease on public websites of China Health Department, CISDCP, Zhejiang Health Department, and Zhejiang Statistical Bureau were collected and understood. Secondly, data were acquired from secondary sources and analyzed anonymously, therefore no participant was required to provide written informed consent.

## 3. Results

### 3.1. General Characteristics of Notifiable Diseases

During the period of 2008–2017, a total of 32 types and 1,994,740 cases of notifiable diseases in children aged 0–14 years, including 266 deaths, were reported in Zhejiang Province, with an annual average morbidity rate of 2502.87/100,000 and an annual average mortality rate of 0.33/100,000. There were no cases and deaths involving plague, cholera, infectious atypical pneumonia, human infection with avian influenza, polio, anthrax, diphtheria and filariasis. No Class A infectious diseases were reported. Twenty-two types and 72,041 cases of Class B infectious diseases were reported, including 138 deaths; 10 types and 1,922,699 cases of Class C infectious diseases were reported, including 128 deaths.

### 3.2. Temporal Distribution

In 2008–2017, morbidity of Class B infectious diseases showed a significant downward trend, from 185.34/100,000 in 2008 to 54.36/100,000 in 2017 (χ^2^_trend_ = 11,093.22, *p* < 0.05), with an annual morbidity of 90.39/100,000; morbidity of Class C infectious diseases showed a fluctuating upward trend, from 1352.97/100,000 in 2008 to 2549.03/100,000 in 2017 (χ^2^_trend_ = 97,595.69, *p* < 0.05), with an average annual morbidity rate of 2412.47/100,000 (Table 1).

The top 5 reported Class B infectious diseases were dysentery, scarlet fever, measles, Influenza A (H1N1) and syphilis. The morbidity of measles, dysentery and syphilis showed a decline (measles: χ^2^_trend_ = 10,156.59, *p* < 0.05; dysentery: χ^2^_trend_ = 6301.75, *p* < 0.05; syphilis: χ^2^_trend_ = 3376.99, *p* < 0.05); and that of scarlet fever was on the rise in recent years (χ^2^_trend_ = 4185.20, *p* < 0.05). Influenza A (H1N1) was classified as a Class B infectious disease in 2009; 5805 cases of influenza A (H1N1) were reported in 2009, ranking first among Class B infectious diseases reported in the same year. This disease showed a decline in 2010 (χ^2^ = 5126.04, *p* < 0.05), and the number of cases reported was between 3 and 259 in 2010–2013. Since 1 January 2014, it was removed from Class B to Class C under the management of existing influenza [4] (Figure 1).

The top 5 reported Class C infectious diseases were hand-foot-and-mouth disease (HFMD), other infectious diarrheal diseases, mumps, influenza and acute hemorrhagic conjunctivitis, among which the morbidity of HFMD, other infectious diarrheal diseases, and influenza were on the rise, while the morbidity of acute hemorrhagic conjunctivitis and mumps were decreasing year by year. In 2010, 11,789 cases of acute hemorrhagic conjunctivitis were reported, and thereafter the number of cases reported decreased rapidly (Figure 2).

### 3.3. Population Distribution

In 2008–2017, a total of 1,216,228 male cases and 778,512 female cases of notifiable infectious diseases were reported, with a male to female ratio of 1.56:1. In terms of the age of onset, the morbidity of infectious diseases was the highest in the 1-year-old group, reaching 9806.26/100,000, which was significantly higher than other age groups. The age of onset was mostly <5 years old. The general trend was that with the increase of age, the number of cases gradually declined. The average annual male morbidity was 2886.98/100,000, and the average annual female morbidity was 2072.16/100,000, the male morbidity higher than the female morbidity (χ^2^ = 54,033.12, *p* < 0.05). The number of male cases in each year was higher than female cases in 2008–2017 (Figure 3).

### 3.4. Geographic Distribution

In 2008–2017, cases were reported in 11 cities throughout the whole province, with the highest reported morbidity in Ningbo (3416.99/100,000), Shaoxing (2912.12/100,000), Huzhou (2757.21/10,000), Jiaxing (2739.87/100,000) and Hangzhou (2664.22/100,000) (Figure 4). The top five cities reported deaths were in Ningbo (42 cases), Jinhua (41 cases), Wenzhou City (39 cases), Taizhou (36 cases), and Shaoxing (26 cases). The top five cities in terms of cases per square kilometer were Jiaxing, Ningbo, Wenzhou, Taizhou and Shaoxing, respectively (Table 2). Cases density and population density were correlated with each other (Pearson’s correlation coefficient = 0.846, *p* = 0.001).

### 3.5. Infectious Diseases by Classification and Transmission Routes

The number of cases of Class C infectious diseases accounted for 96.39% of the total number of cases reported, mainly including HFMD, other infectious diarrhea diseases and mumps, accounting for 89.19% (1,714,925/1,922,699) of the reported cases of Class C infectious diseases. The reported Class B infectious diseases mainly included respiratory infectious diseases and intestinal infectious diseases, accounting for 83.98% (60,499/72,041) of the reported cases of Class B infectious diseases (Table 3). A total of 36,735 cases of respiratory infectious diseases were reported, with an average annual morbidity of 46.09/100,000, including measles, tuberculosis, scarlet fever, influenza A (H1N1), whooping cough, epidemic cerebrospinal meningitis, and human infection with H7N9 avian influenza. There were 6864 cases of measles reported in 2008 and 5805 cases of influenza A (H1N1) reported in 2009 in Zhejiang Province respectively, and the number of respiratory infection cases reported in 2008–2009 accounted for 42.20% of the total number of respiratory cases in the same period (15,504/36,735). A total of 23,764 cases of intestinal infectious diseases were reported, with an average annual morbidity of 29.82/100,000, including dysentery, typhoid and paratyphoid fever, and viral hepatitis (except hepatitis B and hepatitis C). The morbidity rate dropped from 70.09/100,000 in 2008 to 12.28/100,000 in 2017, showing a downward trend on a year by year basis.

### 3.6. Epidemiological Characteristics of Major Infectious Diseases

#### 3.6.1. Hand-Foot-and-Mouth Disease (HFMD)

HFMD is an emerging infectious disease caused by a group of viruses, including enteroviruses, coxsackieviruses, echoviruses, and polioviruses. On 2 May 2008, HFMD was included in Class C notifiable diseases [6]. In 2008–2017, the number of reported cases of HFMD in children aged 0–14 years accounted for 57.15% of all notifiable infectious diseases. The number of reported cases increased rapidly, and the number of cases and the morbidity in 2009–2016 both ranked first among notifiable diseases. A total of 1,139,986 cases were reported, of which 89.34% were reported in the 0~4 years old group; all of the 119 deaths were reported in children aged 0~4 years old. The ratio of male to female morbidity was 1.54:1. HFMD was of the highest annual morbidity among all the reported infectious diseases, reaching 1430.38/100,000, and its annual average mortality rate reached 0.15/100,000, with an annual average case-fatality rate of 0.01% (Table 4).

Since the morbidity rose sharply in early 2008, it retained cyclical fluctuations at a high level (χ^2^_trend_ = 299,306.23, *p* < 0.05), and the morbidity in 2014 was as high as 2849.80/100,000.

Table 4 shows that there were 2040 severe cases, including 119 deaths in 2008–2017. Of these cases, there were 40,716 laboratory diagnosed cases, including 14,032 cases of enterovirus 71 (EV71), 8683 cases of coxsackieviruses A16 (CV-A16), and 18,001 cases of other enteroviruses. From 2008 to 2012, the predominant serotype was EV71 (49.84%), followed by CV-A16 (26.03%) and other enteroviruses (24.13%). In 2013–2017, the predominant serotype was other enteroviruses (55.89%), followed by EV71 (25.52%) and CV-A16 (18.59%).

#### 3.6.2. Other Infectious Diarrheal Diseases

In 2008–2017, a total of 574,939 cases were reported, including 9 deaths. The annual average morbidity was 721.40/100,000, ranking second among all notifiable diseases. The morbidity level fluctuated remarkably and showed a fluctuating upward trend. In 2017, the morbidity reached a peak of 1179.40/100,000.

It showed an overall declining trend with age, and the cases were mainly concentrated in the 0~ and 1~ year-old groups, with a total of 434,464 cases (75.57%) reported. A total of 9 deaths were reported, including 8 deaths in the 0~ year-old group and 1 death in the 1~ year-old group. The ratio of male to female morbidity was 1.57:1 (Figure 5).

#### 3.6.3. Mumps

The average annual morbidity of mumps ranked third among all the notifiable diseases, reaching 168.84/100,000. A total of 134,556 cases were reported, but without death. The morbidity dropped from 264.78/100,000 in 2008 to 77.59/100,000 in 2017, showing a significant downward trend (χ^2^_trend_ = 21,815.20, *p* < 0.001). The ratio of male to female morbidity was 1.84:1.

#### 3.6.4. Influenza, Influenza A (H1N1) and Avian Influenza A (H7N9)

The average annual morbidity of influenza ranked fourth among all the notifiable diseases, reaching 47.40/100,000. A total of 37,775 cases were reported, but without death. In 2017, the morbidity of this disease ranked third among all the notifiable diseases, increasing from 4.87/100,000 in 2008 to 190.24/100,000 in 2017. The morbidity showed a significant upward trend (χ^2^_trend_ = 48,328.69, *p* < 0.05). In 2009, 5805 cases of influenza A (H1N1), including 19 deaths, were reported, ranking first among Class B infectious diseases reported in the same year; this disease showed a rapid decline in 2010, and the number of cases reported in 2010–2013 was between 3 and 259, without death being reported. Avian influenza (H7N9) has been included in the management of Class B infectious diseases since 2013. A total of 4 cases were reported in 2013–2017, with no death; 3 cases were reported in 2014 and 1 case in 2016, and no cases were reported in other years.

#### 3.6.5. Dysentery

The average annual morbidity of this disease ranked the fifth among all the notifiable diseases, reaching 25.61/100,000. A total of 20,413 cases were reported, with one death in 2008. The morbidity dropped from 61.58/100,000 in 2008 to 9.68/100,000 in 2017, showing a significant downward trend (χ^2^_trend_ = 6301.75, *p* < 0.05). The ratio of male to female morbidity was 1.63:1.

#### 3.6.6. Hepatitis B

A total of 2224 cases were reported, accounting for 48.68% (2224/4569) of the total number of viral hepatitis cases. The morbidity dropped from 6.59/100,000 in 2008 to 1.02/100,000 in 2017, showing a downward trend year by year (χ^2^_trend_ = 861.28, *p* < 0.05). The average annual decline was 9.39%. The ratio of male to female morbidity was 1.88:1.

## 4. Discussion

To the best of our knowledge, this is the first population-based study on epidemiological description of notifiable infectious diseases in children aged 0–14 years in Zhejiang Province, China, covering 1,994,740 cases from the CISDCP in the past 10 years.

The surveillance results showed that the average annual morbidity of children aged 0–14 years in Zhejiang Province in 2008–2017 was 2502.87/100,000, which was higher than the annual morbidity of notifiable diseases in the whole population in the same period (709.66/100,000). Analysis of the surveillance report of notifiable diseases in Zhejiang Province in 2008–2017 revealed that, since 2008, no Class A infectious disease had occurred in children aged 0–14 years old; the morbidity of Class B infectious diseases showed a decreasing trend year by year, from 185.34/100,000 in 2008 to 54.36/100,000 in 2017; the morbidity of Class C infectious diseases showed a fluctuating upward trend, from 1352.97/100,000 in 2008 to 2549.03/100,000 in 2017. This surveillance results indicated the harmful effect of Class C infectious diseases is wider for children.

The annual morbidity of typhoid/paratyphoid, hepatitis A, hepatitis E and hepatitis (unclassified) was low (usually < 2/100,000) in Class B intestinal infectious diseases in the last decade. Dysentery was the only Class B infectious disease that ranked among the top 5 infectious diseases, while its morbidity showed a significant downward trend year by year. The surveillance results were similar with the trend of the whole nation [8]. The potential reason for the incidence downward trend may be related to improvements in socio-economic status and the strengthening of public health management.

Measles, tuberculosis, epidemic cerebrospinal meningitis, whooping cough and diphtheria in Class B respiratory diseases have all been included in the national immunization program. The Zhejiang Provincial immunization information system (ZJIIS) was established in 2004 with links to all immunization clinics. The provincial survey in 2014 showed the proportions of children with vaccination cards and registered in ZJIIS were 94.0% and 87.4%, respectively [9]. The high vaccination rate over many years has formed a good immunity barrier. Since 2009, the morbidity of such infectious diseases in children in Zhejiang Province has been relatively low. The previous study [10] reported a measles epidemic in Zhejiang Province in 2008. Afterwards, ten measures (such as emergency immunization, selective supplementary immunization activities, patient management, etc.) for measles elimination were conducted thoroughly to improve population immunity and helped to decrease the incidence, especially for children aged from 8 month to 14 years old [11]. In 2009, the morbidity of measles declined at a clear rate and then remained at a low level in the following years. This finding is consistent with those reported in Poland and Iran [12,13]. Scarlet fever is currently the only Class B respiratory infectious disease without vaccine prevention, ranking first in the morbidity of Class B respiratory infectious diseases in children under 15 years old in Zhejiang Province. In addition, its morbidity has been on the rise in recent years, which is basically consistent with the whole national trend and other countries [14,15,16].

This study also showed that the morbidity of hepatitis B in children under 15 years old in Zhejiang Province dropped from 6.59/100,000 in 2008 to 1.02/100,000 in 2017, showing a downward trend year by year, and a morbidity decline by 9.39% annually. In 1992, Zhejiang Province started promoting the self-funded neonatal hepatitis B vaccination program. In 2002, the hepatitis B vaccine was included in the national free immunization plan for children [17]. It has achieved a certain impact on the prevention and treatment of hepatitis B. A similar finding was reported in other countries [18].

The infectious diseases in children aged 0–14 years in Zhejiang Province were mainly Class C infectious diseases, accounting for 96.39%, which was mainly related to the morbidity of HFMD and other infectious diarrheal diseases. This was consistent with the national epidemic surveillance results [19]. This study showed that HFMD mainly occurred in the ≤4 year-old group, and severe cases and deaths mainly occurred in children under 5 years old. Previous studies also believed that the risk of severe cases and death is significantly higher in younger patients [20,21], suggesting that reducing the prevalence of severe illness and mortality in young children should be the focus of prevention and control. Before 2012, the predominant serotype was EV71; since 2013 other enteroviruses became the predominant serotype. This result of viruses proportions in last five years was similar to the study of Thailand in 2016 [22].

For other infectious diarrheal diseases, immunization is only available against rotavirus. The pathogens are rich in species, and children can have multiple onsets. This study showed that other infectious diarrheal diseases mainly occurred in the 0~ year-old group and the 1~ year-old group, accounting for 75.57%. Deaths mainly occurred in 0~ year-old infants. The potential reasons for this may be related to the following factors. The immune system and intestinal functions of infants and young children are not yet fully developed on the one hand, and it may also be closely related to their own health status and their parents’ poor health-related knowledge and health habits on the other hand [22]. In 2008, the vaccine containing mumps component (the measles-mumps-rubella combination vaccine) was included in the national immunization program. The results of this study showed that the morbidity of mumps in children aged 0–14 years in Zhejiang Province dropped from 264.78/100,000 in 2008 to 77.59/100,000 in 2017, showing a significant downward trend, consistent with the situations in Beijing, Tianjin and Shanghai [23].

The new infectious disease influenza A (H1N1) was globally prevalent in 2009 [24]. On 30 April 2009, it was included in the management of Class B infectious diseases in China [7]. The morbidity of influenza A (H1N1) in Zhejiang Province increased sharply in 2009 [25], this study showed that 5805 cases of influenza A (H1N1) were reported, ranking the first among Class B infectious diseases reported in the same year. As of 1 January 2014, it was included in Class C infectious disease (influenza) statistics [4]. Previous studies suggested that children under the age of 15 in Zhejiang Province are high-risk populations for influenza [26]; this study result also showed that the morbidity of influenza in children aged 0–14 years in Zhejiang Province increased from 4.87/100,000 in 2008 to 190.24/100,000 in 2017. Since the influenza A (H1N1) pandemic, with the strengthened influenza prevention and control work requirements and the expansion of influenza surveillance areas, the reported morbidity of influenza in children under the age of 15 in Zhejiang province has been on the rise. It is recommended that vaccination against influenza and antiviral drugs should be adopted for the prevention and treatment of influenza in children [27], and vaccination is a crucial step in preventing influenza and reducing health care-related infections [28].

After the first case of human infection with avian influenza A (H7N9) was found in China in February 2013 [29], epidemics were reported in the eastern and southeastern coastal provinces including Zhejiang, Shanghai, and Jiangsu [30,31]. These new high-prevalence epidemics caused broad concern across the whole society. Since 1 November 2013, it has been included in the management of Class B infectious diseases. This study showed that 4 cases of avian influenza A (H7N9) were reported in 2013–2017, with no death. This was consistent with others studies, which indicated that H7N9 mild cases were in children, while the severe patients were the adult [32,33].

There was another finding which was that the number of reported cases per square kilometer was directly proportional to population density, reflecting that he high reported number of cases occur in relatively crowded cities. A similar finding was reported in another study [34]. The dense population conditions in urban, poor sanitation in suburb and the high mobility of its floating migrant children may lead to the spread of infectious diseases.

## 5. Limitations

There are a few limitations in this study. First, cases information were acquired from the CISDCP—A passive monitoring system. Some factors including detection capability and availability of health facilities may influence the data quality. The results in this study could be biased by these factors, which induced a lower reported incidence rate per year. According to two surveys by China CDC in 2013 and 2015, both of which were based on a nationally representative sample of about 2000 cases in CISDCP, the rates of underreporting for all notifiable infectious disease were less than 10% [35,36]. Therefore, we thought the quality of the data was acceptable. Second, CISDCP is based on total population, therefore seasonal characteristics of infectious diseases in children aged 0–14 years couldn’t be gained via CISDCP during the study period.

## 6. Conclusions

In summary, there was a downward trend in Class B infectious diseases in Zhejiang Province between 2008 and 2017. In particular, the incidence rates of vaccine-preventable diseases and intestinal infectious diseases kept in low levels in the last decade. However, morbidity of Class C infectious diseases showed a fluctuating upward trend at the same time. We should focus on the prevention and control of HFMD, other infectious diarrheal diseases, mumps and influenza in children aged 0–14 years in Zhejiang Province. It is recommended to strengthen epidemic surveillance, take early prevention and control measures (such as timely vaccination, risk population management, nosocomial infection control, supervision, feedback, and objective assessment), to reduce the incidence rate of infectious diseases in younger children.

## Figures and Tables

**Figure 1 ijerph-16-00168-f001:**
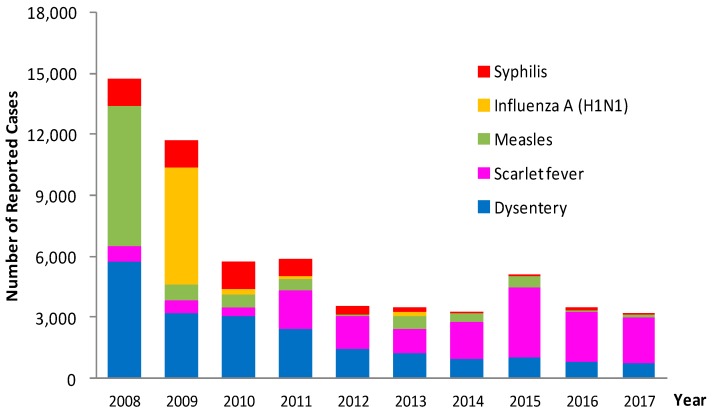
Annual morbidity trend of top 5 Class B infectious diseases in children aged 0–14 years reported in Zhejiang Province, 2008–2017.

**Figure 2 ijerph-16-00168-f002:**
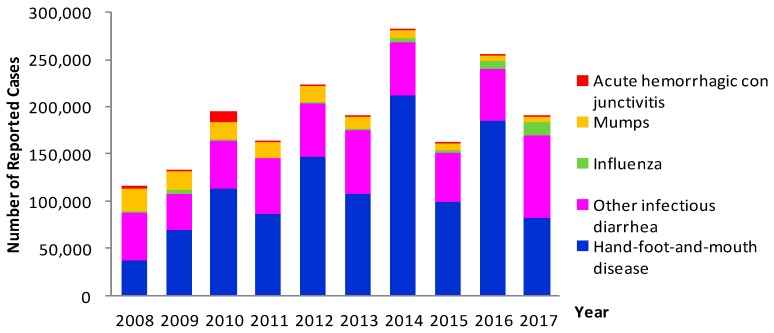
Annual morbidity trend of top 5 Class C infectious diseases in children aged 0–14 years reported in Zhejiang Province, 2008–2017.

**Figure 3 ijerph-16-00168-f003:**
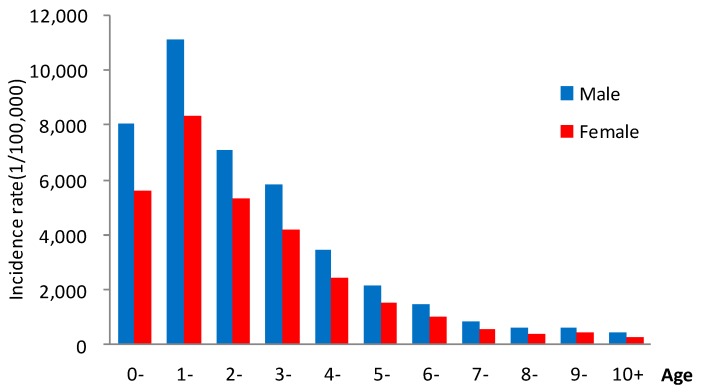
Morbidity of notifiable diseases in all age groups of children aged 0–14 years in Zhejiang Province, 2008–2017.

**Figure 4 ijerph-16-00168-f004:**
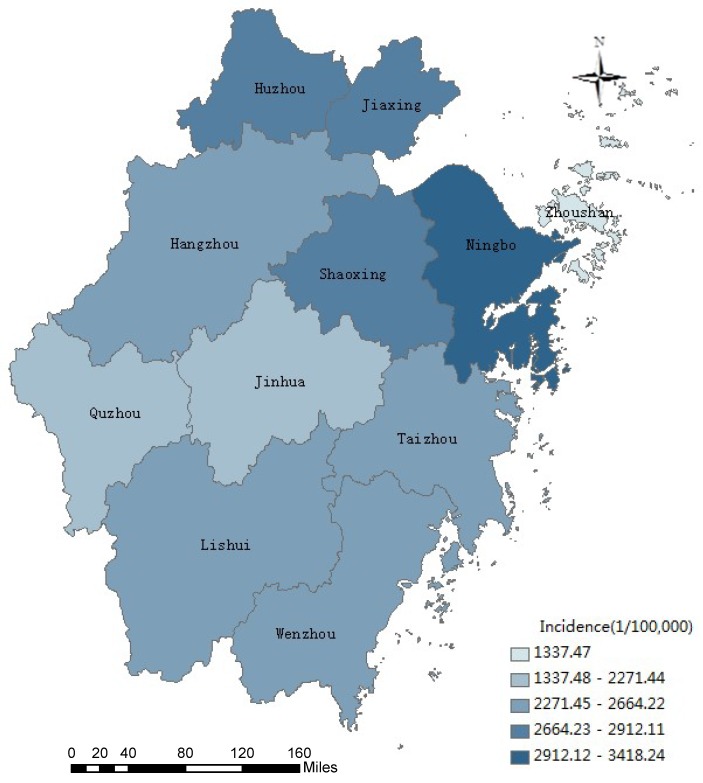
Geographic distribution of annual average incidence rate of notifiable diseases in children aged 0–14 years reported in Zhejiang Province, 2008–2017.

**Figure 5 ijerph-16-00168-f005:**
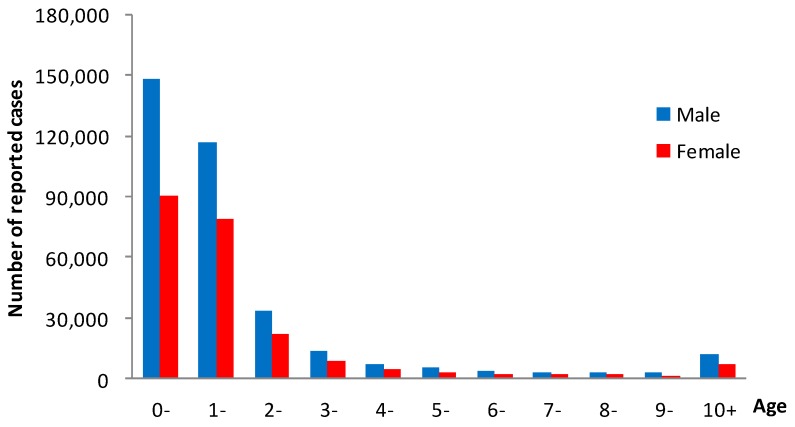
Distribution of age groups of other infectious diarrheal diseases in children aged 0–14 years in Zhejiang Province in 2008–2017.

**Table 1 ijerph-16-00168-t001:** Overall morbidity of notifiable diseases reported in children aged 0–14 years in Zhejiang Province, 2008–2017.

Year	Class B Disease			Class C Disease			Total		
	Number of Diseases	Number of Cases	Morbidity *	Number of Diseases	Number of Cases	Morbidity *	Number of Diseases	Number of Cases	Morbidity *
2008	15	17,193	185.34	6	125,506	1352.97	21	142,699	1538.32
2009	17	13,648	145.52	7	135,251	1442.12	24	148,899	1587.64
2010	17	7244	76.52	6	196,200	2072.5	23	203,444	2149.02
2011	19	7106	98.8	8	166,054	2308.85	27	173,160	2407.65
2012	19	4540	62.51	7	223,294	3074.24	26	227,834	3136.75
2013	18	4211	57.12	7	189,442	2569.74	25	193,653	2626.86
2014	16	3965	53.38	6	280,958	3782.17	22	284,923	3835.55
2015	18	5860	78.69	7	161,219	2164.88	25	167,079	2243.57
2016	17	4224	56.91	6	254,847	3433.46	23	259,071	3490.36
2017	17	4050	54.36	6	189,928	2549.03	23	193,978	2603.38
Total	22	72,041	90.39	10	1,922,699	2412.47	32	1,994,740	2502.87

* Morbidity: 1/100,000.

**Table 2 ijerph-16-00168-t002:** Correlation relationship between population density and reported cases density.

City	Population Density	Reported Cases Density
Jaixing	1178.54	3.82
Ningbo	810.69	3.31
Wenzhou	760.46	2.85
Taizhou	646.05	2.74
Shaoxing	604.17	2.21
Hangzhou	553.63	1.70
Huzhou	511.17	1.65
Jinhua	504.48	1.61
Zhoushan	804.16	1.19
Quzhou	244.43	0.86
Lishui	125.16	0.52

(1) Population density = population/km^2^; (2) Reported cases density = Reported cases/km^2^; (3) km^2^: square kilometer; (4) Pearson’s correlation coefficient = 0.846, *p* = 0.001.

**Table 3 ijerph-16-00168-t003:** Overall morbidity of notifiable diseases in Zhejiang in children aged 0-14 years between 2008 and 2017.

Disease Types	Reported Cases	Constituent Ratio (%)	Total Mortality (/100,000)	Deaths	Total Mortality (/100,000)
Class B infectious diseases	72,041	3.61	90.39	138	0.17
Intestinal infectious diseases	23,764	1.19	29.82	1	0
Respiratory infectious diseases	36,735	1.84	46.09	31	0.04
Blood-borne and sexually transmitted infectious diseases	10,152	0.51	12.74	7	0.01
Natural focal and insect-borne infectious diseases	600	0.03	0.75	25	0.03
Neonatal tetanus	790	0.04	0.99	74	0.09
Class C infectious diseases	1,922,699	96.39	2412.47	128	0.16
Total	1,994,740	100.00	2502.87	266	0.33

Note: Class B infectious diseases were classified into five sub-classes by routes of transmission, etiology or sources of infection, including intestinal infectious diseases, respiratory infectious diseases, blood-borne and sexually transmitted infectious infections, natural focal and insect-borne infectious diseases, and neonatal tetanus.

**Table 4 ijerph-16-00168-t004:** Morbidity of hand-foot-and-mouth disease in Zhejiang in children aged 0–14 years between 2008 and 2017.

Year	Reported Cases	Morbidity *	Deaths	Mortality *	Severe Cases	Laboratory Diagnosed Cases	EV71	CV-A16	Other Enteroviruses
2008	36,570	394.23	7	0.08	164	1455	804	63	588
2009	69,259	738.48	8	0.09	148	1085	461	433	191
2010	113,103	1194.73	37	0.39	1041	4338	2373	1316	649
2011	86,858	1207.69	24	0.33	251	3714	2089	743	882
2012	147,385	2029.15	17	0.23	111	4380	1735	1342	1303
2013	108108	1466.46	3	0.04	44	4383	975	396	3012
2014	211,697	2849.80	13	0.18	152	6619	2172	1491	2956
2015	99,821	1340.42	1	0.01	25	4258	690	813	2755
2016	185,205	2495.20	8	0.11	88	5969	1482	1541	2946
2017	81,980	1100.26	1	0.01	16	4515	1251	545	2719
Total	1,139,986	1430.38	119	0.15	2040	40,716	14,032	8683	18,001

* Morbidity: 1/100,000; Mortality: 1/100,000.

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
