# Peer review of "Analysis of Epidemiological Characteristics of Notifiable Diseases Reported in Children Aged 0–14 Years from 2008 to 2017 in Zhejiang Province, China"

_ijerph, 2019, doi:10.3390/ijerph16020168_

Round 1

Reviewer 1 Report

This is an important and valuable report.

Data completeness: on line 337 you note that passive monitoring is a key limitation. In the US system there is a facility to email or phone those reporting in cases to verify and improve accuracy and completeness. Are data completeness verified and checked in China? 

Interpretation: You comment on high vaccination rates. What was the vaccination status of those with infectious diseases. Are the infections in groups of lower socio economic status who were not vaccinated?

Table 2 is an interesting presentation of population density and disease density? Is the correlation due to socio economic status? 

Is vaccine refusal by parents recorded?

Please report the similarities and differences of your results to similar countries and report insights this provides to interpretation of your data.

Line 207 please comment on whether the viruses you cite for HFMD occur in other countries in the same proportions.

Figure 2 please provide data for the components of "other infections." (a large number of infections).

Typos. Line 111 Please change Person to Pearson.

Line 172 move 3.4 Geographic Distribution from bottom of table to text

Line 173 change high to highest

Author Response

Dear editors and reviewers:

  Thanks very much for the editors and reviewers’ comments concerning our manuscript.

The main corrections and responds to the comments are as follows:

Responds to the reviewers’ comments:

Reviewer #1:

1)Data completeness: on line 337 you note that passive monitoring is a key limitation. In the US system there is a facility to email or phone those reporting in cases to verify and improve accuracy and completeness. Are data completeness verified and checked in China? 

Response: We have revised these in the discussions part in lines 399-402 on page 16.

China CDC does quality survey for CISDCP every few years. According to the recent two surveys in 2013 and 2015, the total infectious disease reporting rate was 91.87% and 95.65%, respectively.

2) Interpretation: You comment on high vaccination rates. What was the vaccination status of those with infectious diseases. Are the infections in groups of lower socio economic status who were not vaccinated?

Response: The Zhejiang provincial immunization imformation system(ZJIIS)was established in 2004 with links to all immunization clinics. According to the data of the provincial survey in 2014, the proportions of children with vaccination cards and registered in ZJIIS were 94.0% and 87.4%, respectively. So the high vaccination over many years has formed a good immune barrier.

Zhejiang is a socio-economically developed province and has attracted more than 20 million migrant people from other areas of China since 1990s. There are almost 200 thousand migrant children lived in Zhejiang; but some migrant children with high frequency of immigration may live in Zhejiang province for a short period of time and receive only 1–2 vaccinations. Consequently, the possible reason for infections was immigration of children.

According to the Vaccination Coverage Survey in 2014 and 2017, the main reason for infections was immigration of children. Low education level of mother or poverty may confer to the result.( Hu, y.; Liang,H,; Wang, Y.;Chen,Y.P. Inequities in Childhood Vaccination Coverage in Zhejiang, Province: Evidence from a Decomposition Analysis on Two-Round Surveys. Int. J.Environ. Res. Public Health. 2018, 15, 2000.)

3) Table 2 is an interesting presentation of population density and disease density? Is the correlation due to socio economic status? 

Response: We have discussed these in the discussions part in lines 391-393 on page 16.

We think the dense population conditions in urban, poor sanitation in suburb and high mobility of migrant children may lead to the spread of infectious diseases.

4)  Is vaccine refusal by parents recorded?

Explain: We didn’t find the reference about vaccine refusal by parents.One study showed the most frequent reason for non-vaccination was parent’s fear of adverse events of immunization. Delayed immunizations were associated with mother having a lower education level, and having a lower household income. (Hu, y.; Li,Q,; Chen,Y.P. Timeliness of Childhood Primary Immunization and Risk Factors Related with Delays: Evidence from the 2014 Zhejiang Provincial Vaccination Coverage Survey. Int. J.Environ. Res. Public Health. 2017, 14, 1086)

5) Please report the similarities and differences of your results to similar countries and report insights this provides to interpretation of your data.

Response: We have discussed these in the discussions part in lines 316-317,331-332,334-336,342-343,391-392,etc.

6) Line 207 please comment on whether the viruses you cite for HFMD occur in other countries in the same proportions.

Response: We have discussed these in the discussions part in lines351-353 on page 15.

In this surveillance result, from 2008 to 2012, the predominant serotype was EV71 (49.84%), and in 2013-2017, the predominant serotype was other enteroviruses (55.89%).

7) Figure 2 please provide data for the components of "other infections." (a large number of infections).

Response: We have added the definition of other infections in method part in lines 111-112 on page 3.

Other infectious diarrhea is a group of infectious diarrhea diseases except cholera, dysentery, typhoid/paratyphoid.

8) Typos. Line 111 Please change Person to Pearson.

Response: We have modified the text in line 141 on page 4.

9)Line 172 move 3.4 Geographic Distribution from bottom of table to text

Response: We have modified the text in line 204 on page 8.

10) Line 173 change high to highest

Response: We have modified the text in line 205 on page 8.

 Special thanks to you for all of your good comments.

Other changes:

We tried our best to improve the manuscript and made some small changes in addition to the above revisions. The small changes do not influence the content and framework of the paper. Therefore, we did not list them here, but we marked all the changes in the revised paper.

We appreciate for the editors and reviewers’ hard work earnestly, and hope that the correction will meet with approval.

Thanks very much for your comments and suggestion again!

Reviewer 2 Report

The authors present a well-written epidemiologic study of notifiable diseases in children up to age 14 in the Zhejiang province of China between 2008-2017. This manuscript was interesting to read, and the figures and tables present the data clearly. I do have some concerns about the manuscript that need to be addressed.

Major comments:

1.      The authors should detail how data on mortality were obtained.

2.     In the results section, the authors make reference to diseases showing a “rapid decline” in annual incidence over the study period. What were criteria for rapid decline? This should be clarified in the manuscript.

3.     Figure 1 nicely displays the number of cases reported of several individual diseases annually. There appears to have been a quite dramatic decrease in the number of reported measles cases between 2008 and 2009. The authors should comment in the discussion on why this decrease was seen.

4.     In Figure 2 and lines 152-157, the authors should define what was included in the category of “other infectious diarrhea.”

5.     The authors should clarify whether morbidity rates by male or female sex presented as per 100,000 total or per 100,000 of that sex category.

6.     The conclusion statement begins with a comment about improvement in living standards and health habits. However, the authors have not presented data to support this statement. They also make statements about children and their tendency toward enteral disease transmission, which extend beyond the data presented. The entire conclusions paragraph overreaches the conclusions that can be drawn from this manuscript, and frankly, the possibilities that may explain the rates of infection in children would be better suited for the discussion section. The concluding statement needs to be completely rewritten.

Minor comments:

1.     The abstract makes reference to Class A, Class B, and Class C infectious diseases. The authors should define what these classifications mean if they are going to refer to them. This would improve readability of the abstract for a general audience.

2.     The flow of ideas in the introduction is not logical. The third paragraph really should be the first paragraph, to grip the reader’s attention and frame the purpose of the analysis presented. The second paragraph of the introduction would be more appropriately placed in the methods section under the Data Resources paragraph.

3.     It is not clear to me why there are two sections for “Data Analysis” and “Statistical Analysis.”

4.     In the General Characteristics of Notifiable Diseases section, the authors begin a sentence with a number, which should be avoided. Please either spell out twenty-two or rephrase the sentence to not begin with a number.

5.     In line 133 on page 4 the authors state that the rate of Class B infections downtrended to 54.36/10. Is this a typo? Should it read /100,000 like the other morbidity measures listed?

6.     The sentence in lines 137-139 is complicated and hard to follow. I recommend splitting this up into multiple sentences rather than using a semicolon.

7.     In line 163 the authors state “On term of age of onset…” but this should read “In terms of age of onset…”

Author Response

Dear editors and reviewers:

  Thanks very much for the editors and reviewers’ comments concerning our manuscript.

The main corrections and responds to the comments are as follows:

Responds to the reviewers’ comments:

Reviewer #2: Major comments:

1)  The authors should detail how data on mortality were obtained.

Response: We have added these in method part in lines 131-133 on page 4.

 The Mortality (M) was calculated by dividing the number of reported deaths (D) via the CISDCP by the number of aged 0-14years inhabitants (I) registered in local public health facilities (M =D/I).

2)In the results section, the authors make reference to diseases showing a “rapid decline” in annual incidence over the study period. What were criteria for rapid decline? This should be clarified in the manuscript.

Response: There were no criteria for rapid decline. We have modified “Rapid” to “obvious” lines170,173 on page 4,5.

3) Figure 1 nicely displays the number of cases reported of several individual diseases annually. There appears to have been a quite dramatic decrease in the number of reported measles cases between 2008 and 2009. The authors should comment in the discussion on why this decrease was seen.

Response: We have discussed these in the discussions part in lines 326-336 on page14.

There was a measles epidemic in Zhejiang Province in 2008. Then ten measures (such as (such as emergency immunization, selective supplementary immunization activities, patient management,etc.)for measles elimination were well conducted to improve the population immunity, and helped to decrease the incidence especially for children aged from 8 month to 14 years old. In 2009, the morbidity of measles declined obviously and then kept in a low level in following years.

4) In Figure 2 and lines 152-157, the authors should define what was included in the category of “other infectious diarrhea.”

Response: We have added the definition of other infections in method part in lines 111-112 on page 3.

Other infectious diarrhea is a group of infectious diarrhea diseases except cholera, dysentery, typhoid/paratyphoid.

5) The authors should clarify whether morbidity rates by male or female sex presented as per 100,000 total or per 100,000 of that sex category.

Response: We have added the definition of other infections in method part in lines 134-135 on page4.

The morbidity rates by male or female sex presented as per 100,000 of that sex category.

6) The conclusion statement begins with a comment about improvement in living standards and health habits. However, the authors have not presented data to support this statement. They also make statements about children and their tendency toward enteral disease transmission, which extend beyond the data presented. The entire conclusions paragraph overreaches the conclusions that can be drawn from this manuscript, and frankly, the possibilities that may explain the rates of infection in children would be better suited for the discussion section. The concluding statement needs to be completely rewritten.

Response:  We have rewrited the conclusion part in lines 407-414 on page 17.

Reviewer #2: Minor comments:

1) The abstract makes reference to Class A, Class B, and Class C infectious diseases. The authors should define what these classifications mean if they are going to refer to them. This would improve readability of the abstract for a general audience.

Response: We have added these in method part in lines 97-100 on page 3.

Thirty-nine notifiable infectious diseases are divided into classes A, B, and C according to their epidemic levels and potential population threats by“Law of the People's Republic of China on the Prevention and Control of Infectious Diseases”. Both Class A and B notifiable diseases are with high risk of outbreak in rapid spread, while  Class C cause less severe epidemics than those of the Class B.

2) The flow of ideas in the introduction is not logical. The third paragraph really should be the first paragraph, to grip the reader’s attention and frame the purpose of the analysis presented. The second paragraph of the introduction would be more appropriately placed in the methods section under the Data Resources paragraph.

Response: We have removed the third paragraph to the first paragraph and modified the second paragraph in lines36-51 on page 1-2.

3) It is not clear to me why there are two sections for “Data Analysis” and “Statistical Analysis.”

Response: We have merged “Statistical Analysis” section into “Data Analysis”section.

4) In the General Characteristics of Notifiable Diseases section, the authors begin a sentence with a number, which should be avoided. Please either spell out twenty-two or rephrase the sentence to not begin with a number.

Response: We have modified the text in line 159 on page 4.

5) In line 133 on page 4 the authors state that the rate of Class B infections downtrended to 54.36/10. Is this a typo? Should it read /100,000 like the other morbidity measures listed?

Response: This is a typo. We have modified the text in line 164 on page 4.

6) The sentence in lines 137-139 is complicated and hard to follow. I recommend splitting this up into multiple sentences rather than using a semicolon.

Response: We have corrected the sentence in lines 169-170 on page 4.

  7) In line 163 the authors state “On term of age of onset…” but this should read “In terms of age of onset…”

Response: We have modified the text in line 194 on page 7.

Special thanks to you for all of your good comments.

Other changes:

We tried our best to improve the manuscript and made some small changes in addition to the above revisions. The small changes do not influence the content and framework of the paper. Therefore, we did not list them here, but we marked all the changes in the revised paper.

We appreciate for the editors and reviewers’ hard work earnestly, and hope that the correction will meet with approval.

Thanks very much for your comments and suggestion again!

Round 2

Reviewer 1 Report

1. You state: "The top five

187 cities reported deaths were in Ningbo (42 cases), Jinhua (41 cases), Wenzhou City (39 cases), Taizhou

188 (36 cases), and Shaoxing (26 cases). The top five cities in terms of cases per square kilometer were

189 Jiaxing, Ningbo, Wenzhou, Taizhou and Shaoxing, respectively (Table 2). Cases density and

190 population density were correlated with each other (Pearson’s correlation coefficient = 0.846, p =

191 0.001)."

Do you have data from these cities about specific residential areas or socio-economic groups or by immigrant status that would enable public health officials to focus on groups at risk and improve health care?

You state: " According to two surveys by

372 ChinaCDC in 2013 and 2015, both of which were based on a nationally representative sample of

373 about 2,000 cases in CISDCP, the rates of underreporting for all notifiable infectious disease were

374 less than 10% [35-36]. So we thought the quality of the data was acceptable."

Do you have data about which types of cases are most underreported and can you suggest interventions to improve reporting?

You state: "384 children aged 0-14 years in Zhejiang Province. It is recommended to strengthen epidemic

385 surveillance, take early prevention and control measures, to reduce the incidence rate of infectious

386 diseases in younger children."

This is vague, Can you please suggest specific interventions for detection and prevention that would be helpful to public health officials? 

Author Response

Dear editors and reviewers:

  Thanks very much for the editors and reviewers’ comments concerning our manuscript.

The main corrections and responds to the comments are as follows:

Responds to the reviewers’ comments:

Reviewer #1:

1)     You state: "The top five cities reported deaths were in Ningbo (42 cases), Jinhua (41 cases), Wenzhou City (39 cases), Taizhou (36 cases), and Shaoxing (26 cases). The top five cities in terms of cases per square kilometer were Jiaxing, Ningbo, Wenzhou, Taizhou and Shaoxing, respectively (Table 2). Cases density and population density were correlated with each other (Pearson’s correlation coefficient = 0.846, p =0.001)."

Do you have data from these cities about specific residential areas or socio-economic groups or by immigrant status that would enable public health officials to focus on groups at risk and improve health care?

Response:  We have added these in the conclusions part in lines 418-420 on page 17.

Ningbo, Wenzhou and Taizhou, located on the southeast coast of Zhejiang Province, are important industrial and port cities. Migrant population accounts for 10% of the total local population. Jiaxing and Shaoxing, located in north-central Zhejiang Province, are important economic regions of the Yangtze River Delta. The population density of Jiaxing is the highest in Zhejiang Province. The migrant population accounts for about 11% in Shaoxing. Jinhua is located in central Zhejiang Province. Yiwu, the largest small commodities trade center in the world, is one of the counties of jinhua district. By 2017, the migrant population accounts for about 40% in Yiwu.

Strategy: It is recommended to strengthen epidemic surveillance, take early prevention and control measures(such as timely vaccination, risk population management, nosocomial infection control, supervision, feedback, and objective assessment), to reduce the incidence rate of infectious diseases in younger children.

2)    You state: " According to two surveys by China CDC in 2013 and 2015, both of which were

based on a nationally representative sample of about 2,000 cases in CISDCP, the rates of underreporting for all notifiable infectious disease were less than 10% [35-36]. So we thought the quality of the data was acceptable."

Do you have data about which types of cases are most underreported and can you suggest interventions to improve reporting?

Response: Two surveys showed that hepatitis B, hepatitis C and syphilis were most underreported. The main reason was HBV, HCV and TPLA must be tested before surgery, while some doctors were very easy to forget to report the positive results. Besides, it was difficult for public health specialists to supervise because most of them have no access to laboratory testing system.

Strategy: With the development of information technology, more and more medical institutions have established and used electronic medical record system. The system have some features including the real-time collection, data exchange and monitoring. Therefore, public health specialists can view outpatient service system,inpatient system and laboratory test results real-timely. The advanced management can improve the quality of infectious diseases reporting effectively.

3)    You state: " children aged 0-14 years in Zhejiang Province. It is recommended to

strengthen epidemic surveillance, take early prevention and control measures, to reduce the incidence rate of infectious diseases in younger children."

This is vague, Can you please suggest specific interventions for detection and prevention that would be helpful to public health officials? 

Response: We have added these in the conclusion part in lines 418-420 on page 17.

Special thanks to you for all of your good comments.

We appreciate for the editors and reviewers’ hard work earnestly, and hope that the correction will meet with approval.

Thanks very much for your comments and suggestion again!

Reviewer 2 Report

The authors present a revised manuscript that addressed most of my prior comments. I have only a few remaining concerns.

1.     The authors have changed the term “rapid decline” to “obvious decline.” However, this misses the point that it must be stated clearly and objectively how such cases were determined. This still needs to be addressed.

2.     The authors have included a statement that “Other infectious diarrhea is a group of infectious diarrhea diseases except cholera, dysentery, typhoid/paratyphoid.” However, this misses the point that not all childhood diarrheal illnesses are reportable, and the specific reportable diseases included in the “other infectious diarrhea” category need to be listed.

3.     There are numerous typos in this revised manuscript that are likely a byproduct of the editing process. Nonetheless, these need to be addressed prior to publication.

Author Response

Dear editors and reviewers:

  Thanks very much for the editors and reviewers’ comments concerning our manuscript.

The main corrections and responds to the comments are as follows:

Responds to the reviewers’ comments:

Reviewer #2:

1)       The authors have changed the term “rapid decline” to “obvious decline.” However, this

misses the point that it must be stated clearly and objectively how such cases were determined. This still needs to be addressed.

Response: We have added statistical tests in lines 174-178 on page 5.

We have done statistical tests about the annual decline of dysentery, measles,syphilis,and influenza A(H1N1),and deleted“rapid” and “obvious”.(measles:χ2trend=10156.59,P< 0.05;dysentery: χ2trend=6301.75, P<0.05; syphilis:χ2trend=3376.99, P<0.05; scarlet fever:χ2trend=4185.20,P< 0.05;influenza A(H1N1):χ2= 5126.04, P < 0.05).

2)       The authors have included a statement that “Other infectious diarrhea is a group of

infectious diarrhea diseases except cholera, dysentery, typhoid/paratyphoid.” However, this misses the point that not all childhood diarrheal illnesses are reportable, and the specific reportable diseases included in the “other infectious diarrhea” category need to be listed.

Response: We have added these in the Materials and Methods part in lines 111-116 on page 3.

According to “Diagnostic criteria for infections diarrhea”(WS 271-2007), enacted by the Ministry of Health of the People’s Republic of China, “other infectious diarrhea”in CISDCP is defined as a group of infectious diarrhea diseases with pathogens including rotavirus,norovirus,salmonella, enteropathogenic escherichia coli(EPEC), enteroinvasive escherichia coli (EIEC), etc, but not including cholera, dysentery, typhoid/paratyphoid.

3)       There are numerous typos in this revised manuscript that are likely a byproduct of the

editing process. Nonetheless, these need to be addressed prior to publication.

Response: We have checked the spelling in the manuscript.

Special thanks to you for all of your good comments.

We appreciate for the editors and reviewers’ hard work earnestly, and hope that the correction will meet with approval.

Thanks very much for your comments and suggestion again!